# Parental developmental screening with CARE: A pilot hybrid assessment and intervention with vulnerable families in Colombia

**Juan Giraldo-Huertas** *

Department of Developmental and Educative Psychology, Universidad de la Sabana, Chía, Colombia

* juangh@unisabana.edu.co

## Abstract

Poverty and scarcity of resources make children in low-and-middle-income countries at risk of not reaching their developmental potential. Despite a near-universal interest in risk reduction, effective interventions like enhancing reading skills in parents to diminish developmental delay remain elusive for the great majority of vulnerable families. We undertook a efficacy study for parental use of a booklet called CARE for developmental screening of children between 36 to 60 months old ($M$ = 44.0, $SD$ = 7.5). All participants ($N$ = 50), lived in vulnerable, low-income neighborhoods in Colombia. The study followed a pilot Quasi-Randomised Control Trial design (i.e., control group participants assigned based on non-random criteria) of parent training with a CARE intervention group compared to a control group. Data was analyzed using two-way ANCOVA for sociodemographic variables' interaction with follow-up results and one-way ANCOVA to evaluate the relations between the intervention and post-measurement of developmental delays and cautions and other language related-skills outcomes, while controlling for pre-measurements. These analyses indicated that the CARE booklet intervention enhanced children's developmental status and narrative skills (developmental screening delay items, $F(1, 47) = 10.45$, $p = .002$, partial $\eta^2 = .182$; narrative devices scores, $F(1, 17) = 4.87$, $p = .041$, partial $\eta^2 = .223$). Several limitations (e.g., sample size) and possible implications for the analysis of children's developmental potential are discussed and considered for future research, along with the effects of the COVID-19 pandemic on the closure of preschools and community care centers.

## Introduction

While progress has recently been made in the early identification of developmental delay in children using screening tools adapted to local cultures [1–4], it remains the case that more than 40% of children under 5 years old in low-income countries are at risk of not reaching their developmental potential because of psychosocial deprivation associated with poverty [5–7]. Effects of poverty and economic deprivation include compromised academic achievement and long-lasting negative effects on general wellbeing [8]. Currently, the COVID-19 pandemic increases the risk to vulnerable children cut off from educational services, which is likely to

**Data Availability Statement:** The final dataset and accompanying code are available on the Open Science Framework, DOI: 10.17605/OSF.IO/JQP8F.

**Funding:** This work was supported the corresponding author dedication by grants awarded by the internal research fund of the Universidad de la Sabana (https://www.unisabana.edu.co/investigacion/fomento-de-la-investigacion/) and, Ministerio de Ciencia, Tecnología e Innovación de Colombia – Minciencias (https://minciencias.gov.co/formacion-de-alto-nivel/apoyo-nacionales-exterior), Grant #860, for Doctoral studies support. Opinions, findings, and conclusions from this report are those of the authors and do not necessarily reflect the views of Minciencias. The funders had no role in study design, data collection and analysis, decision to publish, or preparation of the manuscript.

**Competing interests:** The authors have declared that no competing interests exist.

widen the achievement gap between them and their more economically secure counterparts [9]. Poverty entails not just monetary insecurity. It has a multidimensional impact [10], including health and nutrition deficiency, limited access to nursery or preschool education, and poor material standards of living, including essential services such as electricity, sanitation, and drinking water [11]. Multidimensional analysis of poverty includes a wide consideration of sociodemographic effects on early developmental screening and interventions in lower-income countries [12].

Early developmental screening of children living in low-income homes, followed by programs which encourage parents to support their preschool child's cognitive development, has been shown to produce beneficial effects [4, 13]. Also, family support programs for home literacy practices and engagement in tracking child development milestones have had positive effects on young children when parents are encouraged and trained to support their child's development [14], confirming the moderating effects that families have on the relationship between poverty and child developmental and academic outcomes [15]. However, not all findings have been positive. A longitudinal study recently indicated that after early interventions, positive changes were reported in parenting skills and developmental effects when children were aged 6, 12 and 36 months, but no statistically reliable results were found when same participant children were assessed at 5 years old [16]. Contradictions about the permanence of positive changes in outcomes for at-risk preschool children following parental interventions make imperative the identification of effective early intervention programs that are deliverable in low-resource contexts.

One idea is to use remote interventions based on online solutions. However, many of the population in low- and middle-income countries (LMICs) report no internet connection or access to online services. Before the pandemic, out of the 2.6 billion people living in LMICs, 40% had access to an internet connection, compared to 75% of the people in high-income countries [17]. Technological scarcity in LMICs means that virtual solutions for early learning are insufficient to help most at-risk children. In contrast, home-based routines involving frequent parental monitoring, related to cognitive developmental milestones, can lead to successful interventions that improve the lives of families in extreme poverty or vulnerable conditions [18]. The importance of early reading and playing time with parents is related to routines and habits at home that vary from one socioeconomic level to another, extending deep inequalities in COVID–19 conditions [7, 19]. Likewise, the closure of local daycare and preschool institutions means that most activities carried out in open areas such as parks, that are related to developmental outcomes in children, are diminished and negatively impact families [20]. It is thus necessary to design other remote or hybrid methods that helps to increase home-based parental screening and interaction routines, as an alternative to online, exclusive internet-based solutions. This study aimed to report the effects on children's developmental outcomes when parents used a tool for early developmental screening at home after a hybrid (i.e., in person and remote) intervention.

## A solution for remote screening administrated by parents: The CARE booklet

Early developmental screening (EDS) with well-designed tools has been deployed to assess the effect of intervention programs on specific outcomes [21, 22]. Well-designed EDS tools also reduce financial and time costs for fundamental research and public health activities, such as assessing early developmental status at an individual level [23], including in LMICs [22].

In the last decade, various studies have evaluated EDS tools deployed at primary healthcare services in LMICs [1, 24, 25]. These review studies did not mention any screening tool particularly designed for or used in Colombia [1, 24]. However, Colombia's Ministry of Health uses

the Abbreviated Development Scale (ADS-1; in Spanish, Escala Abreviada del Desarrollo) [26] in various institutional scenarios, including children's centers and public kindergartens around the country. The Ministry of Health presented the ADS-1 with no published information on its conceptualization or pilot testing, nor any complete analysis of validity or reliability [27]. A partial validation analysis of the ADS-1 for the language and hearing domain in 4- to 5-year-old children indicated low predictive ability (sensitivity: 54%, specificity: 42%) and poor agreement with a gold standard for early detection of language and hearing disorders (the Reynell norm-referenced test) on measuring expressive and receptive language skills [28]. We can therefore conclude that to the best of our knowledge, ADS-1 may not be a very suitable tool for the Colombian context, following the standards of Boggs et al. [1]. Moreover, there have been no studies about the "administration of test" effects for this tool, as recommended by several researchers [1, 25]. An administration of test analysis requires comparing caregiver reports with direct child observation. Vitrikas et al. [29] described both a parent-completed EDS tool as an instrument for obtaining screening information through parent participation, and (as a separate instrument) a directly administered EDS tool where information is based on direct observation of the child by a physician or other expert. To probe the potential of an EDS tool to improve the outlook of at-risk children in conditions of poverty in LMICs, the current study describes the use of a parent-administered report of direct observation and activities for children between 36 and 59 months of age. More specifically, it presents the further development of a direct assessment tool conducted by parents and other caregivers called CARE (i.e., Compilation of Activities to Report and Enhance development). We define "parent report" as information obtained from a parent using CARE.

The design of CARE was associated with an intervention program in Colombia called the *Inicio Parejo de la Vida* (IPV; in English, "Equal Start in Life"). The main content of CARE includes activities to report developmental milestones in four domains (personal-social, language and logico-mathematical reasoning, fine motor-adaptive, and gross motor skills). Every item in CARE is closely related to an item in the Haizea-Llevant screening table (HLL) [30], which is a developmental screening version of the Denver Developmental Screening Test (DDST) [31, 32] and the Denver Pre-screening Developmental Questionnaire (PDQ). The selection criteria for the HLL serve two purposes. Firstly, they are intended to allow for easy and rigorous adaptation of developmental screening items in Latin American Spanish through cultural and linguistic modifications obtained from the original Llevant study in the Basque Country [33]. Secondly, the HLL is a widely-used and standardized adaptation of the DDST in various countries, including densely populated areas in Brazil [34] and Colombia [35].

The HLL and DDST items included in CARE and the rest of the design process for the booklet follow the components recommended for the construction and validation of assessment tools used in instrumental screening [36]. Conceptualization and consolidation phases were realized in the IPV program with a sample of 1173 children under 6 years old and their caregivers, in urban settings in two large regions of Colombia (Cundinamarca and Boyacá) [37]. Pilot testing of the reliability and agreement analysis between HLL and CARE indicated that CARE had highly trustworthy diagnostic characteristics (sensitivity: 95%, specificity: 85%) for better and faster classification of an "at risk" status in children aged 24–59 months. Results of a receiver operating curve (ROC) analysis indicated that parental report using CARE is a satisfactory tool for screening diagnostics (Area Under the Curve—AUC: 0.894, trapezoidal Wilcoxon area). All the reliability, agreement and diagnostic-characteristics analysis results were previously carried and published before the intervention and efficacy study [38]. However, it is important to note that using CARE solely as a developmental screening tool, without incorporating its intended characteristics as a parental intervention, does not constitute an actual intervention.

## Parental interventions in LMIC

Two recent meta-analyses of early parenting interventions [39, 40] found medium-sized positive effects on children's early cognitive (Cohen's $d = 0.42$) and linguistic ($d = 0.47$) development. Jeon et al. [39] demonstrated that parent- and family-focused interventions (including psychoeducation, parent- and family-skills training, behavioral and psychosocial interventions) had some benefits for LMIC populations. Twenty-eight of the studies reviewed (88% of all studies included in the metanalysis) showed a significant positive effect of the intervention on a myriad of outcomes, including child and youth mental health and wellbeing, as well as on parenting behaviors and family functioning [39].

However, only a few studies have examined the impact of early parenting interventions on the cognitive performance of children living in poverty, and most of these have reported only limited data about early learning or intervention conditions [41]. Brazil is the sole South American country with an early parenting intervention that focuses on children's cognitive performance. Brasil's program is a parent-child intervention [42] that followed the Reach Out and Read project and the Video Interaction Project [43, 44] with strategies for families under intervention that centered on borrowing children's books on a weekly basis and on reading aloud in monthly parental workshops. Weisleder et al. [42] found that parents in the intervention group engaged in significantly greater cognitive stimulation ($d = 0.43$) and higher quantity and quality of reading interactions ($d = 0.52–0.57$) than controls. Also, at a 9-month follow up, children in the intervention group scored significantly higher than controls on receptive vocabulary ($d = 0.33$), working memory ($d = 0.46$), and IQ ($d = 0.33$).

Likewise, several intervention studies that focused on the development of children's language and communication skills by providing direct training to parents have shown significant benefits in LMIC contexts [45–48]. These included small and medium-sized effects on child expressive ($d = 0.41$) and receptive language ($d = 0.26$). A recent review of 124 studies of language interventions taught to caregivers in homes and classrooms [49] did not find any studies that used one training function with scaffolding or prompting strategies. In our view, the importance of "scaffolding" for parental involvement using tools like CARE and active learning, along with the absence of previous specific interventions with active learning as a training function for caregivers, creates the opportunity for the comprehensive design of a brief parenting intervention using developmental screening reports [50–53].

## The CARE Booklet-Intervention (CBI)

The CARE booklet acts as a guide for parents on how to assess and track their child's developmental progress, and how to provide scaffolding activities to promote their child's development. However, the strict delivery of any screening tool for parent administration, like CARE, is not sufficient as an intervention. A structured intervention that combined the measurement of children in scenarios like children's centers for low-income families in Colombia, with remote protocols for registering parent-child interactions at home, was developed and labeled the CARE Booklet Intervention (CBI). The CARE booklet assessing children's characteristics is a form of home-based records (HBR). An HBR such as CARE, regularly and frequently running in parallel with a full intervention for optimal health and educational systems, is desirable because it can increase parental confidence and engagement in activities with children at home [54]. Parents' engagement in daily activities at home is related to higher-order thinking about their own behavior and how it can enhance, for example, their child's vocabulary development [55].

Also, the CBI could work well at home in cases where interactions with children might be a parental burden. Constant recall and demand for interactions at home might increase parental

stress and the possibility of burnout. Recent studies in the prevalence of parental burnout show that cultural and social values like individualism and the number of hours spent paying attention to children play a significant role in many LMICs, including Colombia [56]. These authors found that specific societal values also played a role: Colombia had the highest scores on the "Indulgence" scale, meaning, according to Roskam et al. [56], that Colombian participants allowed relatively free gratification of basic human drives related to enjoying life and having fun. CBI might transform parental "demand" for interactions with their children into activities reported in the booklet, not for evaluation of parenting, but intended as a support device. Parental support in CBI messages were based on three main theoretical principles: active learning is central to individual child development; social interactions are fundamental to caregiver-child engagement [57]; and development is not an independent process among other processes but, like learning, an aspect of any interactive process in the world [58].

In the CBI, the relevance of parental screening it is not a matter of how much children change and develop but a matter of how, when and with whom they interact. The child might develop in ways that was not anticipated by the report and might not perform according to the kind of development they were expected to show. However, before any hypothesis about indulgence and parental stress or engagement [57], it is necessary to know if the CBI and mediating sociodemographic factors might change children's outcomes, as other parental interventions have been shown to do [49].

The present study reports findings from a pilot quasi-RCT study of a parent-training intervention program for families in poverty with preschool children: the CARE booklet intervention (CBI). Given the generally promising results of parental interventions in LMICs, it is relevant to evaluate the impact of a remote intervention in a small sample of families living in a Spanish-speaking LMIC. This pilot study could be a starting point for more extensive investigation of prompting, guiding, or scaffolding interventions such as the CBI, scaling this to larger samples of participants (parents and caregivers of toddlers and preschool children). We attempted to answer the following research question for our pilot design: Would the CARE booklet intervention (CBI) with parents from vulnerable and scarcity-conditions neighborhoods in Colombia benefit their children's cognitive development and language abilities? As a pilot design, it is necessary to note the exploratory conditions for the current study and be cautious in the analysis of the results.

## Method

The study essentially followed the protocol for a randomized controlled trial (RCT), but the control group was selected based on not attending the randomized session, and therefore the study was classified as a pilot quasi-RCT (Q-RCT) [59]. The trial manager functions were shared between the author and a postgraduate psychology student. A Spanish screening tool known as the Haizea-Llevant observation table (HLL), was used to identify children between 36- to 59-month-old at high risk of not reaching their developmental potential. A member of the research team, blind to group assignment, conducted the assessment. They also employed further standardized measures administered at baseline and post-intervention.

Participants were eligible for the study if they:

1. Had at least one child between 24 and 59 months of age.

2. Had completed the recruitment process and the general developmental screening report (HLL table).

3. Had a classification of "at risk" according to the HLL table, defined as the child presenting ≥1 Delay or ≥2 Cautions.

4. Were able to complete written records in Spanish.

5. Had at least an active phone number, a chat app, or an email address, and to complete an intervention session each week for six weeks and promised to review the intervention materials and answer some questions via phone, chat, or email.

The baseline assessment was conducted if the child met the entry criteria and the parents consented to participate. Prior to the COVID-19 pandemic (July 2019 –February 2020), the screening process identified 268 families (see Fig 1 for the CONSORT flowchart), but 156 families (54.9%) declined to enter the study and were not followed up further. The reasons for their decision were not collected. Thus, 112 families agreed to participate, but 37 of these were excluded for not attending the briefing meetings. This left 75 families who were invited to attend a randomization session, but 27 missed this session. For the remaining 48 families, allocation to either the CARE booklet intervention or another intervention (not reported here) was made from a random numbers table.

Participants for the unreported intervention had no relation to the completed CARE booklet intervention (CBI) or the results from the intervention development process [60]. The unreported intervention and the quasi-RCT protocol helped to prevent the heterogeneity of treatment effect [61], using randomized blocks of six with a computer-generated number sequence that was created a priori on the random.org website. The reason for not reporting the excluded intervention is the limited ability to conduct a comparative analysis with the CBI due to the sample size and an agreement with the independent funding agency to obtain a certain number of participants before creating publications that include the not reported intervention data. An independent research assistant informed participants of their group allocation. The 48 families were randomized to one of two conditions (unreported intervention, $n = 25$; CARE intervention group-only, $n = 23$), and the remaining 27 (those who did not attend the allocation meeting) were assigned to the control group. We cannot establish if this group's lack of attendance the allocation session would affect our results. We assumed that the main reason they did not attend was that they were not available on the meeting date, and there is no reason to think that any material features defining this group would bear on child outcome. The lack of demographic differences between the groups in the final sample ($N = 50$) supports this assumption.

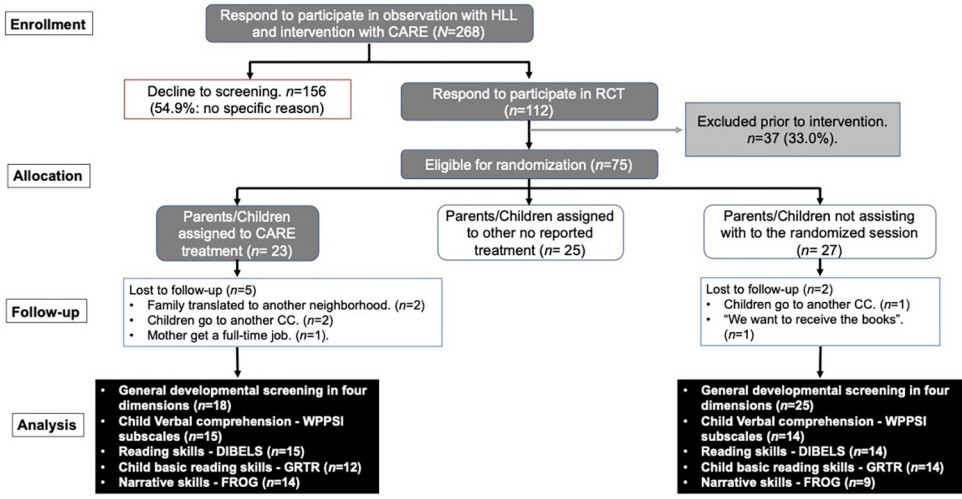

**Fig 1. CARE booklet intervention pilot-study CONSORT flow.**

The study and the intervention involving human participants were reviewed and approved by the board at the Faculty of Psychology of the Universidad de la Sabana, Colombia. The General Directorate of Research at the same university also granted ethical approval for the study (CAG resolution #1517 of 11/19/2015). Permission for data collection was granted in agreement with the legal ruling of Resolution #008430 of 1993 of the Ministry of Health of the Republic of Colombia, which sets out ethical, scientific, technical, and administrative norms for research activity with human participants. Written informed consent from the participant's legal guardian/next of kin was required to participate in this study, following the aforementioned national legislation and institutional requirements.

According to the participant information and preliminary data analysis (see Table 1), all participants fell under the definition of multidimensional poverty [11]. This means that they had low socioeconomic status, with multiple factors affecting them including lower-than-average income on a national scale, health or nutritional deficiency, lack of continuous attendance at nursery or preschool services, and material scarcity in some standards of living (e.g.,

**Table 1. Sociodemographic description of children and mothers in the CBI and Control groups compared with the whole sample.**

|  |  | CBI (*n* = 23) | Control (*n* = 27) | Total (*N* = 50) |
|---|---|---|---|---|
| **Children** |  | *n* (%) | *n* (%) | *n* (%) |
|  | *Gender* |  |  |  |
|  | Girls | 11 (47.8) | 16 (59.3) | 27 (54.0) |
|  | Boys | 12 (52.2) | 11 (40.7) | 23 (46.0) |
|  | *Age group* |  |  |  |
|  | 36–47 months old | 13 (56.5) | 17 (63.0) | 30 (60.0) |
|  | 48–60 months old | 10 (43.5) | 10 (37.0) | 20 (40.0) |
| **Mother** |  |  |  |  |
|  | *Education level* |  |  |  |
|  | Elementary school complete | 1 (4.3) | 2 (7.4) | 3 (6.0) |
|  | Incomplete college | 1 (4.3) | 2 (7.4) | 3 (6.0) |
|  | Complete college or High school | 7 (30.4) | 5 (18.5) | 12 (24.0) |
|  | Apprenticeship certificate or Technician | 5 (21.7) | 10 (37.0) | 15 (30.0) |
|  | Undergraduate degree | 1 (4.3) | 2 (7.4) | 3 (6.0) |
|  | No answer | 8 (34.9) | 6 (22.2) | 14 (28.0) |
|  | *Reading engagement at home* |  |  |  |
|  | Never | 4 (17.4) | 5 (18.5) | 9 (18.0) |
|  | Once or twice per week | 6 (26.0) | 5 (18.5) | 11 (22.0) |
|  | Three to four times per week | 2 (8.7) | 7 (25.9) | 9 (18.0) |
|  | Daily or more per week | 3 (13.0) | 3 (11.1) | 6 (12.0) |
|  | No answer | 8 (34.8) | 7 (25.9) | 15 (30.0) |
|  | *Maternal Employment* |  |  |  |
|  | Working mother | 12 (52.2) | 12 (44.4) | 24 (48.0) |
|  | Non-working mother | 3 (13.0) | 8 (29.6) | 11 (22.0) |
|  | No answer | 8 (34.8) | 7 (25.9) | 15 (30.0) |
|  | *Socioeconomic national scale[+]* |  |  |  |
|  | Very low: Less than 4.5 USD by day | 4 (17.4) | 6 (22.2) | 10 (20.0) |
|  | Low: More than 4.5 USD but less than 10.0 USD by day | 11 (47.8) | 9 (33.3) | 20 (40.0) |
|  | Moderate low: More than 10.0 USD but less than 20.0 USD by day | 0 | 5 (18.5) | 5 (10.0) |
|  | No answer | 8 (34.8) | 7 (25.9) | 15 (30.0) |

Notes: [+]Various sources lead to an approximate calculation of these levels [63–64].

essential home utilities). Initial analysis using the Mann-Whitney test (S1 Table) showed no significant differences between groups in various sociodemographic factors (i.e, children's age in months and gender, mother's education level, reading engagement, mother's employment, and socioeconomic status). Assessments were conducted in-person at a local Children's Centre (CC) for baseline and again not more than two weeks after the last session. Even though in-person assessment was conducted at the CC, the CBI is considered a hybrid intervention with a mix of moderated (i.e., the engagement monitoring via phone call, chat, or email message) and unmoderated procedures (i.e., using at home the CARE booklet) to obtain and deliver related information [62].

## Procedure

The training program was delivered to the parent who identified as the principal caregiver of their child with two in-person sessions at Children's Centre (CC) and other sessions delivered via phone call, chat, and email message. The CARE booklet intervention (CBI) promoted the use of a printed booklet for written reports of different activities at home in four developmental dimensions commonly used for early screening. The intervention involved two in-person parent meetings in small groups (4–6 persons) and receiving instruction from a trained facilitator over one-hour sessions. The dosage per session was less than 50 minutes, to fit in with the high number of other scheduled activities in the CC. One facilitator, a final year-undergraduate student of Psychology at Universidad de la Sabana, oversaw the intervention. CBI required the use of a CARE booklet and materials delivered by the facilitator to parents in an initial session. Each other weekly session, was informed and delivered via phone call, chat, and email message. The remote weekly sessions were about specific activities presented for parents to carry out and report using their CARE booklet. The parental training emphasized the use of CARE to obtain information about daily interactions between parents/caregivers and children, and how the CARE booklet could help them to enhance their children's development through regular monitoring of these interactions at home. The CARE booklet had a section for every activity with an area for daily marking, below instructions for how to report interactions daily for four weeks (see Fig 2). Each intervention session involved a slide presentation, delivered via chat or email message, that focused on a particular dimension of development (personal-

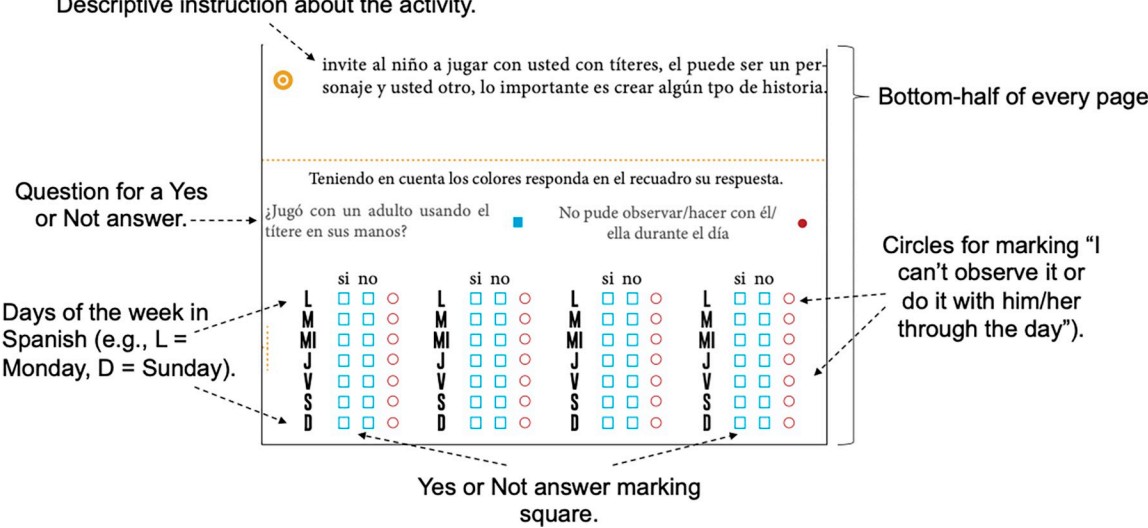

**Fig 2. Area for daily marking in every activity of CARE booklet.**

**Table 2. Session by week description of the CARE booklet intervention and corresponded modality of participation.**

| Session order | CARE intervention content by session | Modality |
|---|---|---|
| Session 1 | Introduction: Explaining the benefits of using the CARE booklet and the importance of establishing an interaction and report routine. | In-person at CC |
| Session 2 | Socialization activities: Review of activities for socialization dimension. | Remote via phone call, chat, and email message |
| Session 3 | Language and logical-mathematical reasoning: Review of activities for language and logical-mathematical reasoning. | Remote via phone call, chat, and email message |
| Session 4 | Object manipulation activities: Review of activities for object fine motor manipulation. | Remote via phone call, chat, and email message |
| Session 5 | Postural development activities: Review of activities for postural and gross motor development | Remote via phone call, chat, and email message |
| Session 6 | Socio-cognitive development in context: Review and highlight the importance of (a) Household and daily activities, like playing musical instruments, painting, and writing, playing in open spaces; (b) Relationship with others; (c) Use of numbers in daily activities, (d) Geo-spatial orientation in daily activities. | In-person at CC |

Note: CC: Children Center.

social, language and logico-mathematical reasoning, fine motor skills, and gross motor skills). Embedded within each session chat or email message were brief video clips of local Spanish-speaking parents demonstrating good practice of the session content. The primary goal of the CBI was not to oblige parents to take part in constant marking or reporting. The act of engaging in interactions with children was more important and was clearly emphasized in every weekly session via phone call, chat, or email message. High frequency and quality of time-shared playing together and performing the activities was the explicit goal.

At the second session (see Table 2), and each subsequent remote session, the facilitator checked via phone call, chat, or email message if that week's daily records had been used, and if any difficulties had been encountered. It was not possible to determine any significant changes week by week and include in the results analysis the daily reporting of activities at home because caregivers' written, and verbal accounts were frequently inconsistent in the registration area of each booklet.

## Description of input from the Children's Centre local services

All the children participating in the study received 70% of their daily nutritional requirements (breakfast, two snacks, and lunch) in the Children's Centre (CC). Additionally, before the COVID-19 pandemic, all children ($N = 50$) attended regular daycare program activities (e.g., singing nursery rhymes, painting). Every parent-child participant dyad in both groups (CBI and Control) could receive a mixture of services in the CC delivered by undergraduate interns (e.g., physiotherapy, general health advice), supervised by professional specialists. One of these services for all participants included reading-aloud practice with children, which was mandatory in the CC since its management was exploring a new structured curriculum of preschool activities.

The parents in the control group only received the instruction to "write or sketch" in a blank paper booklet any activity that they had carried out with their children that month. This control procedure can be classed as a placebo [65]. The control group did not receive any phone call, chat or email message related to the procedures, just one contact before the post-trial measurements when the blank booklet was collected.

## Outcome measures

The outcomes were assessed exclusively in Spanish at a post-intervention assessment conducted 7–8 weeks following the baseline assessment. The assessments were conducted in-person at the Children's Centre by two native Spanish-speaking graduate psychologists who were blind to group membership and trained in the specific measures used. The full set of assessments took 60–90 minutes. The following primary and secondary outcomes were administered at baseline and follow-up using the official or standardized version in Spanish.

### Primary outcomes

a. *General developmental screening report (Haizea-Llevant screening table)*. As previously described, the Haizea-Llevant screening table (HLL) is a developmental screening version of the Denver Developmental Screening Test [31, 32] and the Denver Pre-screening Developmental Questionnaire (PDQ). The HLL for the current study's age range has 57 items and four dimensions: i) Fine motor (12 items); ii) Gross motor (9 items); iii) Language and logico-mathematical reasoning (22 items); and iv) Personal-Social (14 items). For individual assessment classification, the developmental performance score was defined as the number of age-appropriate test items of a dimension that a child successfully passed. For nominal classification, a "Caution" was counted when an age-appropriated item was not passed. If the child was older than the limit age for the 95% of population passing the item, and did not pass it, the item was counted as a "Delay". The counting of caution and delayed items is used to determine risk or developmental delay status [29]. For nominal classification, children with ≥1 Delay or ≥2 Cautions were classified as "At risk". Scoring no Delays and just one Caution was not considered to be indicative of risk. For general screening and analysis of development across different dimensions, both Delays and Cautions were counted.

b. *Child verbal comprehension*. We used the Wechsler Preschool & Primary Scale of Intelligence–Fourth Edition (WPPSI-IV) [66], specifically, the verbal comprehension full scale for ages 2;6–3;11 years, comprising the Receptive Vocabulary, Information, and Picture Naming subscales.

### Secondary outcomes

a. *Literacy skills*: Fluency was assessed using the Dynamic Indicators of Basic Early Literacy Skills (DIBELS, 8th Edition). The DIBELS is a battery of brief fluency measures that can be used for universal screening and progress monitoring in preschool contexts [67]. Letter Naming Fluency (LNF), Phonemic Segmentation Fluency (PSF), and Word Reading Fluency (WRF) were assessed.

b. *Child basic reading skills*: The *Get Ready to Read*! (GRTR) screening measure, which is supported by the US National Center for Learning Disabilities [68], was used. This is a brief, user-friendly measure that assesses children's print knowledge, letter knowledge, and early reading skills. It has established validity in indexing emergent literacy skills in preschool classrooms [68, 69].

c. Narrative skills were measured using the Frog Story (*Frog, where are you*?) [70]. This is a 24-picture story book without words. Botting's procedure [71] was followed, with the child asked to look at every page of the book and then to tell the story. The Frog Story's analysis also followed Botting and included narrative structure (i.e., formal opening, orientation to

characters and setting, explicit mention of the theme, resolution, and formal ending), length (counting both number of words and number of propositions), and use of narrative devices (i.e., mentalizing terms, negatives, causatives, hedges, and words spoken by a character).

Child age and gender, principal caregiver's education level, reported reading engagement at home, mother's employment status, and SES (using Colombian household stratification statistics) were used as covariates.

## Data analysis

All statistical analyses were conducted using IBM SPSS Statistics for Macintosh, Version 25.0 [72]. Due to the small sample in each group of participants ($< 30$) and the nature of the research as a pilot design study [73], as a first step in analysis, we conducted two-way ANCOVAs for a parametric check of interactions between factors (i.e., CBI vs Control) and sociodemographic variables (i.e., child age, gender, principal caregiver's education level, reported reading engagement at home, mother's employment status, and socioeconomic stratification) controlled by pre-intervention scores on each assessment. Then, after the two-way ANCOVA procedure, the interactions (group * sociodemographic variable) were removed, and a one-way ANCOVA was run with only main effects for intervention groups and pre-intervention scores, including all the measurements for primary and secondary outcomes, along with the Cautions and Delay items in the four developmental domains observed in the primary outcome measurements. The unequal sample size in each group prevented an exhaustive pairwise analysis to complete the two-way and one-way ANCOVAs, including post-hoc procedures such as Scheffé's test to find out which pairs of means showed significant differences [74, 75]. Instead of the exhaustive pairwise analysis, the effect size is reported following the principles and recommendations of different experts [76, 77] for interpreting eta-squared ($\eta^2$) and partial eta-squared ($\eta^2{}_{p}$) values as expressions of the amount of variance accounted for by one or more independent variables. The following rules of thumb are used to interpret values for $\eta^2$ and $\eta^2{}_{p}$: $> .01$ = small effect size, $> .06$ = medium effect size, $> .14$ = large effect size.

## Results

### Initial and follow-up group outcomes

Group mean and standard deviation (SD) scores for all variables at the initial and follow-up assessments are shown in Table 3. Overall, primary outcomes presented positive score changes in the CBI group, with fewer Delays and Cautions in the general post-screening, and high indices in the WPPSI-IV subtest post-measurements (i.e., Receptive Vocabulary, Information, and Picture Naming). The control group also showed positive changes, except for Delays (higher in post-screening) and lower values in the WPPSI-IV Picture Naming subtest post-measurement compared with the pre-tested score. Likewise, overall changes were observed in all secondary outcome post-test scores, with one exception: the CBI group had lower scores in the basic reading skills post-test results with *Get Ready to Read*! (GRTR) compared with the pre-test values.

### Sociodemographic effects on pre-post assessments for CBI and Control Group

Two-way ANCOVA indicated statistically significant interactions between the CBI intervention and both child gender and family socioeconomic status, on post-test measurements of HLL Caution items and on the WPPSI-IV Picture Naming subtest, controlling for pre-test measurement values (see Table 4). Likewise, child gender, mother's education level and reading engagement at home had moderating effects on secondary outcomes (the DIBELS Letter

**Table 3. Group means and standard deviations (SD) of scores at initial and interim follow-up assessments for primary and secondary outcomes.**

| Group | CBI | | Control (Only CC services) | |
|---|---|---|---|---|
| | N = 23 | | N = 27 | |
| | **Initial** | **Follow-up** | **Initial** | **Follow-up** |
| Age in months: mean (SD) | 46.6 (6.64) | 47.3 (6.50) | 45.2 (6.90) | 46.7 (7.04) |
| Primary Outcomes: | | | | |
| General screening HLL: Items in Delay | 2.04 (1.69) | 1.04 (1.52) | 1.67 (1.52) | 2.37 (1.47) |
| General screening HLL: Items in Caution | 3.26 (2.38) | 1.96 (2.06) | 3.30 (2.95) | 2.33 (2.17) |
| Child Verbal comprehension: WPPSI-IV Receptive Vocabulary | 9.18 (5.25) | 13.0 (3.84) | 11.3 (5.08) | 13.2 (5.31) |
| Child Verbal comprehension: WPPSI-IV Information | 8.00 (4.52) | 11.5 (4.32) | 12.1 (3.43) | 12.6 (3.30) |
| Child Verbal comprehension: WPPSI-IV Picture naming | 7.95 (4.29) | 10.9 (3.01) | 11.8 (2.65) | 9.86 (4.83) |
| Secondary Outcomes: | | | | |
| DIBELS: Letter Naming Fluency (LNF) | 0.00 (0.00) | 0.07 (0.26) | 0.00 (0.00) | 0.07 (0.27) |
| DIBELS: Phonemic Segmentation Fluency (PSF) | 0.00 (0.00) | 0.00 (0.00) | 0.00 (0.00) | 0.00 (0.00) |
| DIBELS: Word Reading Fluency (WRF) | 0.57 (1.43) | 10.4 (19.4) | 3.67 (7.66) | 12.07 (20.3) |
| Child basic reading skills: GRTR–Get ready to read items | 9.00 (3.20) | 8.50 (2.71) | 7.0 (4.42) | 7.50 (4.91) |
| Frog Story: Structure. | 25.3 (16.0) | 37.1 (22.0) | 22.9 (17.3) | 35.6 (26.0) |
| Frog Story: Length. | 64.7 (71.6) | 159.3 (113.9) | 96.1 (73.6) | 115.4 (72.2) |
| Frog Story: Narrative devices. | 1.33 (3.15) | 7.71 (8.17) | 4.21 (5.07) | 4.89 (4.46) |

Note: CBI: CARE booklet intervention; CC: Children's Centre; HLL: Haizea-Llevant.

Naming Fluency subscale and the *Get Ready To Read*! items) when the CBI group was compared with the control group. The measurements without a significant moderating effect of sociodemographic variables (i.e., HLL Delay items, the WPPSI-IV Receptive Vocabulary and Information subtests, the DIBELS Phonemic Segmentation and Word Reading Fluency subscales, and the Frog Story structure, length, and narrative devices scores) were analyzed to obtain the main effects of the CARE intervention compared to the control group in post-test measurements, controlling for pre-test measurement values.

## Primary outcomes comparison in pre-post assessments for CBI and Control Group

One-way ANCOVAs were conducted to compare the effectiveness of the CARE Booklet Intervention (CBI) in the post-test measurements, controlling for pre-test measurement values. Levene's test and normality checks were carried out and the assumptions were met. There was a

**Table 4. Two-way ANCOVAs results for sociodemographic significative interactions with post measurements by CBI or Control condition and premeasurements values.**

| Co-variable[+] | Measurement (DV) | *df* interaction | *df* Error | Adj. $R^2$ | F | *p* | $\eta^2_p$ |
|---|---|---|---|---|---|---|---|
| Primary Outcomes: | | | | | | | |
| Child: gender | General screening HLL: Items in Caution | 1 | 45 | .464 | 4.18 | .047 | .085 |
| SES | WPPSI-IV Picture Naming | 1 | 15 | .261 | 6.47 | .022 | .301 |
| Secondary Outcomes: | | | | | | | |
| Child: gender | Letter Naming Fluency (LNF) | 1 | 23 | .522 | 5.22 | .032 | .185 |
| Mother's Education level | Letter Naming Fluency (LNF) | 3 | 13 | .320 | 4.12 | .029 | .488 |
| Reading engagement at home | GRTR–Get ready to read items | 3 | 11 | .638 | 4.16 | .034 | .532 |

Notes: [+]Co-variable in interaction with "intervention*Premeasurement".

DV = Dependent variable: Post measurements; HLL = Haizea-Llevant; SES = Socioeconomic status.

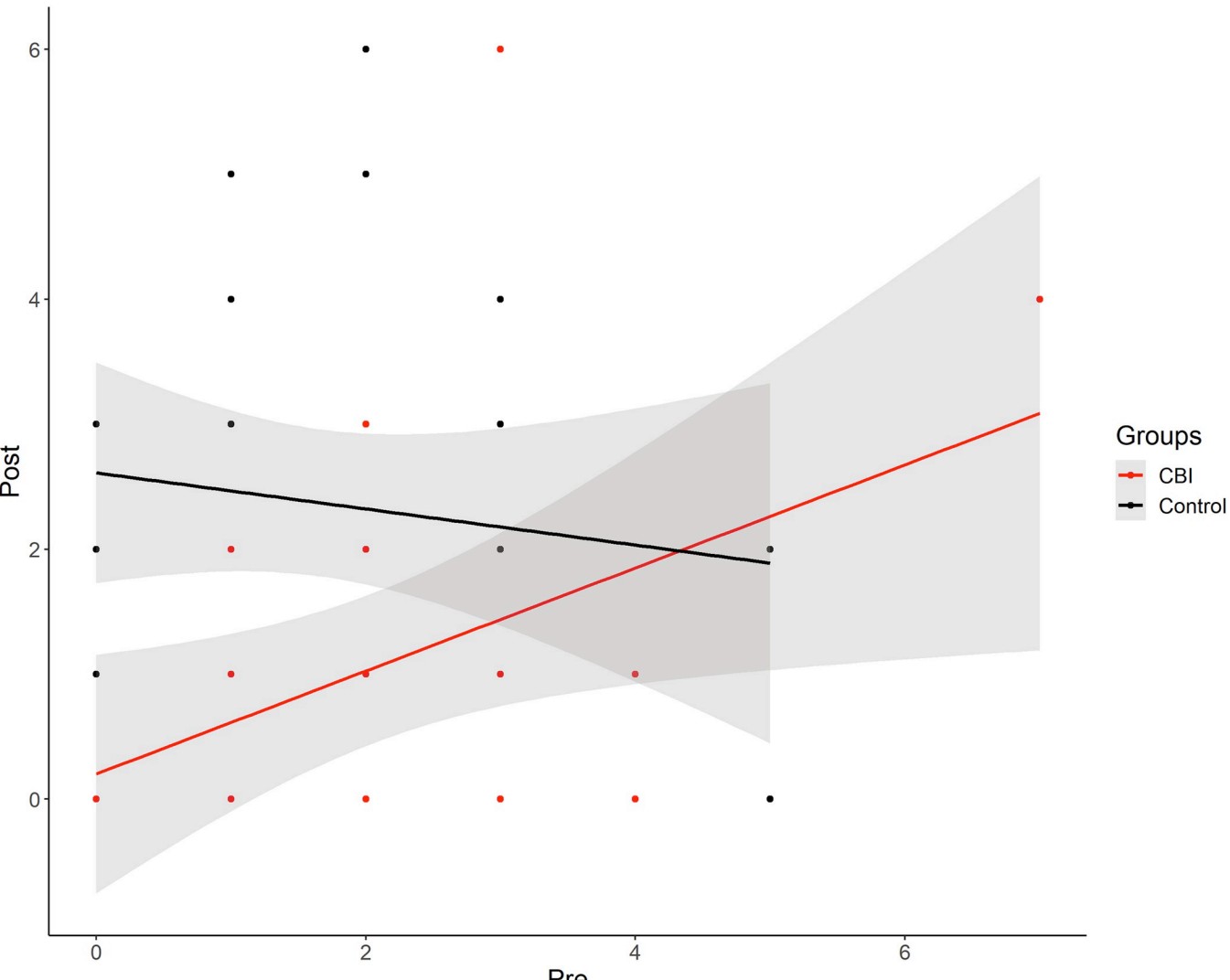

**Fig 3. Scatterplot with regression lines for interactions between pre and post Delays in Haizea-Llevant for CARE booklet intervention (CBI) and Control groups. Note.** This Figure demonstrates a positive effect of the CBI intervention in diminishing the post Delays. The solid lines in colour represent the regression slopes for the CBI (Red) group. The black line corresponds with the regression slope in the Control group and indicates a negative relation for the pre and post Delays. The grey shadow areas for each line represent the standard error for each trendline data.

significant difference between the CBI and control groups on the HLL Delay items, $F(1, 47) =$ 10.45, $p = .002$, partial $\eta^2 = .182$. No differences were found in other primary measurements when CBI and control groups were compared. ANCOVA results in CBI vs Control comparison indicated that mean post-test Delay items in HLL observations differed between the parental training condition and the control group, with a positive association between the pre-test and post-test counts of Delays. That is, children in the CBI condition had an adjusted post-test mean Delay count that was significantly higher than the mean for the control group (see Fig 3).

## Secondary outcomes comparison in pre-post assessments for CBI and Control Group

One-way ANCOVAs were conducted to compare the effectiveness of the CARE Booklet Intervention (CBI) in the post-test measurements, controlling for pre-test measurement values.

**Table 5. Group mean and standard deviation (SD) for initial and follow-up delays and cautions in developmental dimensions in Haizea-Llevant assessment.**

| | CARE booklet intervention | | Control (Only CC services) | |
| | $n = 23$ | | $n = 27$ | |
| Developmental Dimension | Initial | Follow-up | Initial | Follow-up |
| --- | --- | --- | --- | --- |
| Personal-Social: Delays | 0.43 (0.51) | 0.30 (0.56) | 0.96 (0.76) | 0.85 (0.86) |
| Cautions | 1.00 (0.90) | 0.52 (0.73) | 1.30 (0.99) | 0.89 (1.01) |
| Language and logico-mathematical reasoning: Delays | 0.52 (0.67) | 0.43 (0.73) | 0.11 (0.32) | 0.56 (0.64) |
| Cautions | 1.17 (1.34) | 0.74 (1.05) | 1.30 (1.59) | 0.85 (1.23) |
| Fine motor-adaptive: Delays | 0.57 (0.84) | 0.09 (0.29) | 0.30 (0.67) | 0.56 (0.58) |
| Cautions | 0.52 (0.79) | 0.35 (0.65) | 0.19 (0.56) | 0.22 (0.42) |
| Gross motor: Delays | 0.52 (0.67) | 0.22 (0.60) | 0.30 (0.54) | 0.41 (0.57) |
| Cautions | 0.57 (0.66) | 0.35 (0.57) | 0.52 (0.70) | 0.37 (0.56) |

Levene's test and normality checks were carried out and the assumptions were met. There was a significant difference between the CBI and control groups on the Frog Story's narrative devices scores, $F(1, 17) = 4.87$, $p = .041$, partial $\eta^2 = .223$. No differences were found in other secondary measurements when CBI and control groups were compared.

## General screening (Haizea-Llevant) developmental dimensions comparison in pre-post assessments for CBI and Control Group

Group mean and standard deviation (*SD*) scores for the four developmental dimensions in Haizea-Llevant (HLL) screening at the initial and follow-up assessments are shown in Table 5. Overall, primary outcomes presented positive improvements in the CBI group, with fewer Delays and Cautions in all four HLL dimensions.

One-way ANCOVAs were conducted to compare the effectiveness of CBI in post-test measurements for the four developmental dimensions in the HLL screening, while controlling for pre-test measurement values. Levene's test and normality checks were carried out. There was a significant difference between the CBI and control groups in HLL fine motor-adaptive Delays, $F(1, 47) = 10.8$, $p = .002$, partial $\eta^2 = .187$, in favor of the CBI group. No differences were found in other dimensions when the CBI and control groups were compared.

## Effect sizes correction in significant one-way ANCOVA

The ANCOVA procedure in SPSS by default calculates effect size using partial eta squared ($\eta^2_p$). The $\eta^2_p$ statistic belongs to the *r* family of effect sizes [76], and describes the proportion of variance that is explained by group membership (e.g., for bivariate correlations, $r = 0.5$ indicates that 25% of the variance in one variable is explained by the variance in another variable). Several sources [77, 78] recommend the use of corrections for bias, even if corrections do not always lead to a completely unbiased effect size estimation. In the *r* family of effect sizes, the correction for eta squared ($\eta^2$) is known as omega squared ($\omega^2$).

We calculated the $\omega^2$ statistic for the significant ANCOVA results in our study using the following formula:

$$\omega^2 = \frac{SS'_B - (K - 1)MS'_W}{SS'_T + MS'_W}$$

where $SS'_B$ is the sums of square for the adjusted treatment (independent variable), $K-1$ represents the between-groups degrees of freedom, $MS'_W$ is the error mean square and $SS'_T$ is the total sum of squares, all reported by default in the SPSS output for the ANCOVA procedure.

**Table 6. Interpretation of corrected partial η² for ANCOVAs in Care booklet intervention (CBI) vs Control comparisons.**

| Reported significant ANCOVA comparison | | |
|---|---|---|
| CBI vs Control group | $\omega^2$ | Cohen (1988) benchmarks' range effects interpretation |
| All Delay items in Haizea-Llevant | 0.07 | Medium |
| FROG's narrative Devices | 0.06 | Medium |
| Fine motor-adaptive dimension Delays (HLL) | 0.11 | Medium |

Note. HLL = Haizea-llevant.

The resulting $\omega^2$ calculation in reports of significant comparisons between CBI and control groups are shown in Table 6. The $\omega^2$ ranges in value from 0 to 1 and is interpreted as the proportion of total variance in the dependent variable accounted for by the independent variable, partially controlling for the effect of the covariate. Cohen [79] provided benchmarks applied to $\omega^2$ when partial $\eta^2$ is corrected: small effects > .01, medium effects > .06, and large effects > .14.

As can be seen from Table 6, significant effect sizes were evident for CBI with significant medium-sized effects for three measured items.

## Discussion

This study was a pilot Q-RCT of parent training for a remote booklet intervention with pre-school children at risk of not reaching their developmental potential. Findings in the CARE booklet intervention (CBI) were in line with our prediction that the parental training interventions would enhance children's developmental status and language-related skills. Also, follow-up testing on the CBI group revealed significant positive results in both primary and secondary outcomes, compared to the control group that had received local services only. This includes the effect of the decrease in HLL Delay items for the post-test measurements in the CBI group compared with the control group. In the ANCOVAs overall effects interpretation, medium-sized effects were found in one developmental dimension.

The consideration of sociodemographic variables in the two-way ANCOVA analysis is informative for any future RCT, which requires a deep evaluation of home conditions and local services relating to the formal introduction to writing, alongside training in cognitive skills and performance with narrative devices. The results for the CBI group support the screening tool's use as a practical approach to parental reporting that gives confidence in our ability to evaluate activities and interactions at home. Reduction in overall delays, and a medium-sized effect on narrative devices and fine motor-adaptive skills, confirm the CBI as a feasible solution to provide prompts, guidance, and scaffolding to parents and caregivers [49]. However, two-way ANCOVA analysis should also consider the factors of child gender, socio-economic status, mother's education levels and the frequency of reading engagement at home, which all had interaction effects on the developmental language and reading differences between children of parents receiving the CBI compared with children in the control group.

Dosage intervention has had significant effects on previous parental training studies [46], implying a chance to have a more positive impact on children if the dosage of our intervention is increased. Indeed, considering that improvements took place in comparison with a control group who regularly received center-based nutritional services and daycare activities, the results for the CBI intervention are remarkable. On the other hand, effects reported for general language and mathematical milestone items are not consistent with meta-analytical findings for parental intervention results in psychosocial stimulation interventions [39, 40]. These meta-analyses found that the impact of the intervention dosage had clear and robust

moderation effects on child language outcomes, with low dosage associated with a minimal impact. The CBI lasts for six weekly sessions of less than one hour each, suggesting that it is a feasible program for vulnerable low-income families.

The COVID-19 pandemic had causal effects on mental health and caregiving environments, particularly in contexts of poverty. These effects have been observed in Colombia and other LMICs, and have increased the risk of children not reaching their developmental potential [12]. The CBI might help parents in poverty to recognize and change patterns of interaction that are robustly associated with individual trajectories and dynamics of cognitive development [50, 80, 81]. According to Bronfenbrenner's ecological model of development [82] as applied to family and health research [83], the CBI is an intervention designed for microsystems (e.g., daily activities and reading interactions between children and caregivers). Our analysis of the microsystem dynamics relating to the CBI had focused on promoting engagement with sensitive and responsible parenting, but indicated that distinct barriers (e.g., employment and health-related challenges) and facilitators (e.g., knowing other parents in the group, interest in the program topics) had a significant effect in recent studies with socioeconomically disadvantaged families [84], none of which were explored in the current CBI study. However, the consistency and quality of the daily-based information recorded in the CARE booklet were problematic, which made it challenging to identify significant changes in microsystems level by time. Although this is a common issue in research that relies on self-reported data, it is important to acknowledge and address potential sources of bias or errors in the data collection process. In future studies using the CARE booklet, alternative data collection methods or strategies to improve the quality and consistency of the recorded information may prove useful. Additionally, the CBI is a child development intervention supported by current models of early childhood recommended in scalable programs to impact early learning [85, 86], specifically in terms of encouraging dedicated caregiving from parents [87]. It is also a low-cost intervention, compared to others that make exclusive use of home visits. The advantage of a remote intervention compared with those that only use home visits is not only monetary: high parental engagement with home visits is associated with particular maternal characteristics (higher IQ, older mothers, mothers who were employed during pregnancy, mothers with greater knowledge of infant development, and mothers with more positive parenting beliefs), rather than other characteristics (young, unemployed, and/or less well-educated mothers) that are more frequently found in LMICs [88].

## Limitations and future research

All effect sizes in a pilot study should be interpreted with caution because there are several limitations due to the small sample size. A larger sample size is needed for a rigorous and detailed mapping of the effects of similar procedures on poor, low-income families in LMICs, including adequate controls for sociodemographic variation for future post hoc analysis after the two-way ANCOVA procedures to properly evaluate the CBI.

Comparing the CBI with Colombian center-based interventions like the aeioTu program [89] might not be appropriate because it requires a control group with no local service allocation. However, the CBI is a suitable alternative to consider when comparing the child-to-adult dyadic ratio and the length of the intervention. To make a formal comparison between our parent-based intervention and a center-based intervention like that reported by Nores et al. [89], a comprehensive analysis of the scientific literature relating to interventions in Spanish-speaking and LMIC conditions is necessary, but no such study has been conducted so far.

Families allocated to the control (local services only) group were assigned to the control condition because they did not present themselves on the day of randomized allocation and

cannot be considered a truly random control group. However, this reflects a real-life clinical and research situation. Future RCT studies should be completely coherent with randomized allocation and sufficient sample size to avoid potential biases and to increase statistical power to generalize the differences between groups detected.

Another limitation of our study is related to our measure of development. The Haizea-Lle-vant observation table (HLL) is meant as a screening tool and not a diagnostic tool. A high number of Delay and Caution items in the HLL developmental dimensions should not be strictly interpreted as a delay relative to benchmarks in comparison with children of the same age. Therefore, the results were framed in terms of children being "at risk" of a loss of developmental potential, rather than as a diagnosis of a specific developmental delay.

The generalizability of our findings may be affected by using a sample from a community-based services program. Participation in the nutritional program offered in the participating childcare center was voluntary, and parents who had concerns about their child's development may have been more likely to stay in the program than parents who found their children to be on track. This could have inflated our estimates of the prevalence of risk for developmental delay. However, our sample had relatively similar proportions of various sociodemographic risk factors as the broader group of participants previously seen as at risk of developmental delay [14, 41], and for these reasons, it seems reasonable to assume that our sample does not substantially overestimate the risk for developmental delay in this low-income LMIC population.

A major limitation of this study is the lack of direct observational assessment of parent skills, such as caregiver social play sensitivity, which has been shown to produce significant positive changes in infants' cognitive and socioemotional outcomes in previous parental and language interventions [47, 90]. The absence of such assessment was due to scarce financial resources for the Q-RCT. Additionally, levels of commitment of parental involvement (CPI) were not assessed in previous stages of recruitment. A meta-analysis [57] reported that most studies indicate how parents face environmental and personal challenges to participate actively and concluded that levels of engagement are better understood when including a consideration of parental involvement issue. Measures of CPI fall into three main categories: global participation levels, specific participation behaviors, and completion of tasks. The average completion rate for all sessions analyzed by Haine-Schlagel and Escobar-Walsh [57] was 49%, with a range of 19% to 89%. One possible explanation for the low engagement in optional activities for the present study (54.9% at first call) could be attributed to the sociodemographic characteristics that influenced the opportunity of adherence for participants. Comparing our sample and experimental procedure with an RCT of a play-assisted intervention for children living with foster families in extreme poverty [4], two characteristics could affect the CPI in our case. First, families tended not to cancel visits in the Worku et al. intervention [4]. In contrast, our families received all assessments by arriving at the childcare center (CC) and interventions via chat or email message at home. Second, Worku and colleagues' intervention [4] was conducted in a foster family program (SOS-villages), where children lived with assigned foster families and were always cared for by an SOS-village mother or "aunt". In future, direct measures of parental behavior and competence need to be included to demonstrate objective, generalizable difficulties in CPI.

The present study has highlighted significant practical and methodological challenges in conducting an RCT of a remote intervention for parents in low-income conditions with preschool children at risk of not reaching their developmental potential. One major obstacle in our study was the high rate of declining to join after the initial call, which affected our recruitment rate—more than half of the potential families who could have received the intervention (54.9%) opted not to participate. Unfortunately, the reasons for their decision were not

collected. Recent meta-analyses and evidence-based studies [91] report that failure to inform on attrition rates and reasons for attrition can impact the integrity of interventions in LMICs, even in home visit intervention designs [92]. To ensure the quality of future studies, researchers should not only collect information about attrition and non-participation but also incorporate principles and practices that align with new forms of services offered in public health initiatives, such as family-focused and community-based approaches [93].

Moreover, future studies aiming to improve interaction skills between caregivers and children [49] may consider using a low-cost and widely applicable booklet, such as the CARE booklet intervention, to guide parental scaffolding. Our general findings on various developmental screening and language competence variables suggest the need for further work on approaches to parental training in privileged intersubjective spaces for the promotion of children early cognitive skills [94]. This should include RCTs with sufficient sample size and methodological rigor to confirm and extend these tentative findings, and to more clearly demonstrate whether parent training with scaffolding guides has specific beneficial effects on relevant skills and competences of children at risk. One type of intervention includes screening or early developmental monitoring [18, 85] to accomplish remote assessment and intervention in the most vulnerable populations of LMIC.

## Supporting information

**S1 Table.**
(BIN)

## Acknowledgments

The author expresses sincere gratitude to the families and mothers at Amiguitos Royal CC in Bogotá, as well as the internship students of the Pasantía de Cuidado y Desarrollo from the Department of Psychology at the Universidad de la Sabana. Special thanks are extended to Dr. Graham Schafer and Prof. Peter Cooper for their valuable comments on the initial research design, as well as their contributions towards the improvement and clarification of the study during its early stages.

## Author Contributions

**Conceptualization:** Juan Giraldo-Huertas.

**Methodology:** Juan Giraldo-Huertas.

**Project administration:** Juan Giraldo-Huertas.

**Supervision:** Juan Giraldo-Huertas.

**Writing – original draft:** Juan Giraldo-Huertas.

**Writing – review & editing:** Juan Giraldo-Huertas.

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
