## [Decision Letter · Decision Letter 0]

27 Jul 2022

PONE-D-21-40928Parental developmental screening with CARE: a remote assessment and intervention with vulnerable Spanish-speaking familiesPLOS ONE

Dear Dr. Giraldo-Huertas,

Thank you for submitting your manuscript to PLOS ONE. After careful consideration, we feel that it has merit but does not fully meet PLOS ONE’s publication criteria as it currently stands. Therefore, we invite you to submit a revised version of the manuscript that addresses the points raised during the review process. Please note that we have only been able to secure a single reviewer to assess your manuscript. We are issuing a decision on your manuscript at this point to prevent further delays in the evaluation of your manuscript. Please be aware that the editor who handles your revised manuscript might find it necessary to invite additional reviewers to assess this work once the revised manuscript is submitted. However, we will aim to proceed on the basis of this single review if possible. The reviewer has raised a number of concerns that you should respond to as part of your revisions. Please ensure that you provide further detail on the intervention not reported here and why the results of this have not been reported.

In addition to the reviewer’s requests, please clarify the aims of the study in your Introduction. The work is noted as a pilot study – please outline the specific research questions you aimed to address by performing the work.

A significant limitation of the work is the assignment of the control condition. Please mention this concern in your Abstract. Please also note your work as a pilot study in your Title.

We look forward to receiving your revised manuscript.

Kind regards,

George Vousden

Staff Editor

PLOS ONE

Journal Requirements:

Reviewers' comments:

Reviewer's Responses to Questions

**Comments to the Author**

1. Is the manuscript technically sound, and do the data support the conclusions?

Reviewer #1: Yes

2. Has the statistical analysis been performed appropriately and rigorously? 

Reviewer #1: Yes

3. Have the authors made all data underlying the findings in their manuscript fully available?

Reviewer #1: No

4. Is the manuscript presented in an intelligible fashion and written in standard English?

Reviewer #1: No

5. Review Comments to the Author

Reviewer #1: In this manuscript, the authors investigate findings from a pilot quasi-randomized controlled trial of a parenting intervention that combined parent education sessions regarding activities associated with developmental stimulation, and parental monitoring through a home screening booklet (CARE) among families of low socioeconomic status in a LMIC. They found that children whose caregivers were assigned to the intervention screened for fewer developmental delays post intervention than children in the control group. They also found interactions in effects with child gender, maternal education, and family socioeconomic status.

These findings are of potential interest, are important for continued research in child development in LMIC, and provide support for a larger study of this intervention.

However, I have several concerns about the paper, which I have outlined below:

Introduction

1. In the Introduction, the authors note that “only a limited number of studies have examined the impact of early parenting interventions on the cognitive performance of children living in poverty; and most of them have reported only limited data about early learning or intervention conditions.” The authors should include studies by Mendelsohn and colleagues of a reading aloud intervention in Brazil that addresses these issues:

Weisleder, A., Mazzuchelli, D. S., Lopez, A. S., Neto, W. D., Cates, C. B., Gonçalves, H. A., ... & Mendelsohn, A. L. (2018). Reading aloud and child development: a cluster-randomized trial in Brazil. Pediatrics, 141(1).

Mendelsohn, A. L., da Rosa Piccolo, L., Oliveira, J. B. A., Mazzuchelli, D. S., Lopez, A. S., Cates, C. B., & Weisleder, A. (2020). RCT of a reading aloud intervention in Brazil: Do impacts differ depending on parent literacy?. Early Childhood Research Quarterly, 53, 601-611.

Methods

2. Consistency in the name of the intervention would be helpful. Although the authors call the intervention CBI throughout most of the manuscript, they return to CARE several times (e.g., pp. 10, para. 2).

3. The authors indicate that there was another intervention that was not included in the present analyses, but use the acronym for that intervention in the manuscript (DBS; pps. 11 & 25). Naming this intervention from the outset, with a brief description and indication of why it is not included in the present analyses would be a better way to handle this.

4. The manuscript indicates that CBI was a remote intervention, but this was not discussed in the description of the program in the Methods section. Method of delivery should be included there.

5. It would strengthen the manuscript to include comparison statistics between the CBI and Control groups (t-tests or chi-squares) on the descriptive statistics in Table 1. This would provide further evidence that the groups are comparable, even though assignment was done through a quasi-random procedure.

Results

6. The results section would benefit from additional details about the interactions found in the two-way ANCOVAs. Were effects greater for girls vs. boys, mothers with higher vs. lower education, higher vs. lower SES? This would provide important information for planning a larger study.

7. I would also consider presenting the two-way ANCOVA findings after the main effects findings from the one-way ANCOVA, since the main effects are the primary question of interest.

8. The results may be easier to follow if primary and secondary outcomes were separated, with results for the dimension so the HLL screening included with the primary outcome findings.

9. Figure 3 seems to indicate that children with more pre-test delays also had more post-test delays in the CBI group, but not in the Control group. Given that delays increased in the Control, but not the CBI, group, interpretation of this is difficult. Does this indicate that additional children developed delays in the Control group? A different figure may better illustrate the point of the analysis.

Discussion

10. The discussion of parent engagement is interesting, but seems out of place in the limitations section. The manuscript would be stronger if this was included in the Introduction and general Discussion, along with a limitation noting the possibly low rate of up-take and inability to measure what parents actually did at home, since written and verbal indicators did not match.

Overall

11. A minor concern relates to clarity and grammaticality throughout the manuscript. The article would benefit from a thorough review.

6. PLOS authors have the option to publish the peer review history of their article (what does this mean?). If published, this will include your full peer review and any attached files.

Reviewer #1: No

---

## [Author Response · Author response to Decision Letter 0]

22 Oct 2022

Reviewers' comments:

Reviewer's Responses to Questions

Comments to the Author

1. Is the manuscript technically sound, and do the data support the conclusions?

Reviewer #1: Yes

2. Has the statistical analysis been performed appropriately and rigorously?

Reviewer #1: Yes

3. Have the authors made all data underlying the findings in their manuscript fully available?

Reviewer #1: No

4. Is the manuscript presented in an intelligible fashion and written in standard English?

Reviewer #1: No

Reply/action: A revision was made before resubmission by a native English speaker, a highly qualified University associated professor living in Colombia. 

5. Review Comments to the Author

Reviewer #1: In this manuscript, the authors investigate findings from a pilot quasi-randomized controlled trial of a parenting intervention that combined parent education sessions regarding activities associated with developmental stimulation, and parental monitoring through a home screening booklet (CARE) among families of low socioeconomic status in a LMIC. They found that children whose caregivers were assigned to the intervention screened for fewer developmental delays post intervention than children in the control group. They also found interactions in effects with child gender, maternal education, and family socioeconomic status.

These findings are of potential interest, are important for continued research in child development in LMIC, and provide support for a larger study of this intervention.

However, I have several concerns about the paper, which I have outlined below:

Introduction

1. In the Introduction, the authors note that “only a limited number of studies have examined the impact of early parenting interventions on the cognitive performance of children living in poverty; and most of them have reported only limited data about early learning or intervention conditions.” The authors should include studies by Mendelsohn and colleagues of a reading aloud intervention in Brazil that addresses these issues:

Weisleder, A., Mazzuchelli, D. S., Lopez, A. S., Neto, W. D., Cates, C. B., Gonçalves, H. A., ... & Mendelsohn, A. L. (2018). Reading aloud and child development: a cluster-randomized trial in Brazil. Pediatrics, 141(1).

Mendelsohn, A. L., da Rosa Piccolo, L., Oliveira, J. B. A., Mazzuchelli, D. S., Lopez, A. S., Cates, C. B., & Weisleder, A. (2020). RCT of a reading aloud intervention in Brazil: Do impacts differ depending on parent literacy?. Early Childhood Research Quarterly, 53, 601-611.

Reply/action: Included in page 8 of the manuscript

Brazil is the only South American country with an early parenting interventions intervention interested in children’s the cognitive performance (Weisleder et al., 2018). The early intervention reported in Brazil is a parent-child programme (Weisleder et al., 2018) followed the Reach Out and Read and the Video Interaction Project (Mendelsohn et al., 2018, 2020) with strategies for families under intervention, borrowing children’s books on a weekly basis and in focusing on reading aloud in monthly parent workshops. Weisleder et al. (2018) found that parents in the intervention group engaged in significantly greater cognitive stimulation (d = 0.43) and higher quantity and quality of reading interactions (d = 0.52–0.57) than controls. Also, at a 9-month follow up, children in the intervention group scored significantly higher than controls on receptive vocabulary (d = 0.33), working memory (d = 0.46), and IQ (d = 0.33).

Methods

2. Consistency in the name of the intervention would be helpful. Although the authors call the intervention CBI throughout most of the manuscript, they return to CARE several times (e.g., pp. 10, para. 2).

Reply/action: Included in several pages where inconsistencies was detected.

3. The authors indicate that there was another intervention that was not included in the present analyses, but use the acronym for that intervention in the manuscript (DBS; pps. 11 & 25). Naming this intervention from the outset, with a brief description and indication of why it is not included in the present analyses would be a better way to handle this.

Reply/action: as mentioned before, information is extended in pages 11-12 for major clarity.

Participants for the not reported intervention had no information or effect on the completed CARE booklet intervention (CBI) or the results from the intervention development process (Duncan et al., 2020). The not reported intervention and the quasi-RCT protocol help to prevent the heterogeneity of treatment effect (Averitt et al., 2020), using randomized blocks of six with a computer-generated number sequence that was created a priori on the random.org website. The reason for not reporting the excluded intervention is limitations for feasible and comparative analysis with the CBI due to the sample size and the agreement with the independent founding agency for the other intervention to obtain a significant number of participants before publications, including their data. An independent research assistant informed participants of their group allocation. The 48 families were randomized to one of two conditions (not reported intervention, n = 25; CARE intervention group-only, n = 23), and the remaining 27 (those who did not attend the allocation meeting) were assigned to the control group. We cannot establish if they did not attend the allocation session should affect our results. Consequently, we assumed the main reason they did not attend was that they were not available on the meeting date, and there is no reason to assume that any material features defining this group would bear on child outcome. The lack of demographic differences between the groups in the final sample (N = 50) supports this assumption.

4. The manuscript indicates that CBI was a remote intervention, but this was not discussed in the description of the program in the Methods section. Method of delivery should be included there.

Reply/action: information is extended in pages 13-14 for major clarity.

According to the participant information and preliminary data analysis (see Table 1), all participants fell under the definition of multidimensional poverty (Aguilar & Sumner, 2020). This means that they had low socioeconomic status, with multiple factors affecting them including lower-than-average income on a national scale, health or nutritional deficiency, lack of continuous attendance at nursery or preschool services, and material scarcity in some standards of living (e.g., essential home utilities). Initial analysis using the Mann-Whitney test (Supplementary Table 1) showed no significant differences between groups in various sociodemographic factors (i.e, children’s age in months, mother’s education level, reading engagement, mother’s employment, and socioeconomic status). CBI participants were asked to have at least an active phone number, a chat app or an email address, and to complete an intervention session each week for six weeks. Also, the participants promised to review the intervention materials and answer some questions via phone, chat or email. Assessments were conducted at a local Children’s Centre (CC) for baseline and again not more than two weeks after the last session. Even though face-to-face assessment was conducted at the CC, the CBI is considered a remote intervention with a mix of moderated (i.e., the engagement monitoring via phone call, chat or email message) and unmoderated procedures (i.e., the delivered CARE booklet) to obtain and deliver related information (Amso, Cusack, Oakes, Tsuji, & Kirkham, 2022).

5. It would strengthen the manuscript to include comparison statistics between the CBI and Control groups (t-tests or chi-squares) on the descriptive statistics in Table 1. This would provide further evidence that the groups are comparable, even though assignment was done through a quasi-random procedure.

Reply/action: recommendation included in page 14.

Initial analysis using the Mann-Whitney test (Supplementary Table 1) showed no significant differences between groups in various sociodemographic factors (i.e, children’s age in months, mother’s education level, reading engagement, mother’s employment, and socioeconomic status).

Results

6. The results section would benefit from additional details about the interactions found in the two-way ANCOVAs. Were effects greater for girls vs. boys, mothers with higher vs. lower education, higher vs. lower SES? This would provide important information for planning a larger study.

Reply/action: recommendation implied an adjustment in p. 21 with two more technical references:

Due to the small sample, as a first step in analysis, we conducted two-way ANCOVAs for a parametric check of interactions between factors (i.e., CBI vs Control) and sociodemographic variables (i.e., child age, gender, principal caregiver’s education level, reported reading engagement at home, mother’s employment status, and socioeconomic stratification) controlled by pre-intervention scores on each assessment. Then, after the two-way ANCOVA procedure, the interaction (i.e., group * sociodemographic variable) was removed, and the ANCOVA (i.e., one way) was rerun with only main effects for intervention groups and pre-intervention scores, including all the measurements (i.e., primary and secondary outcomes) and the Cautions and Delay items in the four developmental domains observed in the primary outcome measurements. The unequal sample size in each group prevents the exhaustive pairwise analysis to complete the two-way and one-way ANCOVAs, including post-hoc procedures, like the Scheffé’s test, to find out which pairs of means are significant (Westfall, 1997; Westfall & Young, 1993). Instead of the exhaustive pairwise analysis, the effect size is reported following the principles and recommendations of Lakens (2013) and Pek & Flora (2018) for eta-squared (η2) and partial eta-squared (η2p) interpretation, that express the amount of variance accounted for by one or more independent variables. The following rules of thumb are used to interpret values for η2 and η2p: .01= small effect size, .06 = medium effect size, .14 or higher = large effect size

7. I would also consider presenting the two-way ANCOVA findings after the main effects findings from the one-way ANCOVA, since the main effects are the primary question of interest.

Reply/action: we follow the recommendation with the clarifying adjustment in p. 21 described before.

8. The results may be easier to follow if primary and secondary outcomes were separated, with results for the dimension so the HLL screening included with the primary outcome findings.

Reply/action: we follow the recommendation with the following adjustments in p. 25 and 26.

Primary outcomes comparison in pre-post assessments for CBI and Control Group

One-way ANCOVAs were conducted to compare the effectiveness of the CARE intervention (CBI) in the post-test measurements whilst controlling for pre-test measurements values. Levene’s test and normality checks were carried out and the assumptions were met. There was a significant difference between the CBI and control groups on the HLL Delay items, F(1, 47) = 10.45, p = .002, partial η2 = .182. No differences were found in other primary measurements when CBI and control groups were compared. ANCOVA results in CBI vs Control comparison indicate that mean post-tested Delay ítems in HLL observations differed between the parental training conditions and the control group, with a positive association between the pre-test and post-test counts of Delays. Children in CBI conditions displayed adjusted post-test Delays means that were higher than the mean for the control group. Fig 3 below shows the significant nature of the association for the CBI compared to the Control group.

Secondary outcomes comparison in pre-post assessments for CBI and Control Group

One-way ANCOVAs were conducted to compare the effectiveness of the CARE intervention (CBI) in the post-test measurements whilst controlling for pre-test measurements values. Levene’s test and normality checks were carried out and the assumptions were met. There was a significant difference between the CBI and control groups on the Frog Story’s narrative devices scores, F(1, 17) = 4.87, p = .041, partial η2 = .223. No differences were found in other secondary measurements when CBI and control groups were compared. 

9. Figure 3 seems to indicate that children with more pre-test delays also had more post-test delays in the CBI group, but not in the Control group. Given that delays increased in the Control, but not the CBI, group, interpretation of this is difficult. Does this indicate that additional children developed delays in the Control group? A different figure may better illustrate the point of the analysis.

- Reply/action: we follow the recommendation with a new Figure with R (ggplot2) and the following adjustments as Note in p. 26:

Note. This Figure demonstrates a positive effect of the CBI intervention in diminishing the post Delays. The solid lines in colour represent the regression slopes for the CBI (Violet) group. The black line corresponds with the regression slope in the Control group and indicates a negative relation for the pre and post Delays. The grey shadow areas for each line represent the standard error for each trendline data. 

Discussion

10. The discussion of parent engagement is interesting, but seems out of place in the limitations section. The manuscript would be stronger if this was included in the Introduction and general Discussion, along with a limitation noting the possibly low rate of up-take and inability to measure what parents actually did at home, since written and verbal indicators did not match.

Reply/action: we follow the recommendation with the following changes:

In Introduction, p. 9-11:

The CARE Booklet-Intervention (CBI)

CARE acts as a guide for parents on how they can assess and track their child’s developmental progress, and how they can provide scaffolding activities to promote their child’s development. However, the strict delivery of any screening tool for parent administration, like CARE, is not sufficient as an intervention. A structured that combined the opportunity to measure children in scenarios like children centres for low-income families in Colombia and remote protocols for register parent-child interactions at home, helps to give structure to the CARE booklet-intervention (CBI). The CARE booklet assessing children’s characteristics is like home-based records (HBRs). An HBR such CARE regular and frequently running in parallel with a CBI intervention for optimal health and educational systems is not only desirable (Mahadevan & Broaddus-Shea, 2020): it could increase confidence and engagement in parental activities at home. The parental engagement for daily activities at home are related in higher order thinking about their own behaviour and how it can enhance, for example, their child’s vocabulary development (Teepe et al., 2019).

Also, the CBI could work well at home when interactions might be a parental task or burden. Constant recall and demand for interactions at home might increase parental stress and the possibility of parental burnout. Recent studies in the prevalence of parental burnout (Roskam et al., 2021) show that cultural and social values like individualism and number of hours spent paying attention to children play a significant role in many LMICs, including Colombia. Specific societal values also play a role in Roskam and colleagues’ research. Colombia had the highest scores on the “Indulgence” scale, meaning, according to the researchers, that participants (N = 95) allow relatively free gratification of basic and natural human drives related to enjoying life and having fun. CBI might transform parental “demand” for interactions with their children into activities reported in the booklet, not for evaluation of parenting, but intended as a support device. Parent support in CBI messages had three principal interpretations: active learning is central to individual child development; social interactions are fundamental to caregiver-child engagement (Haine-Schlagel & Escobar-Walsh, 2015); and (paraphrasing Lave, 1988) development is not an independent process among other processes but, like learning, it is an aspect of any interactive process in the world. 

In the CBI, the relevance of parental screening it is not a matter of how much children change and develop but a matter of how, when and with whom they interact. The child might develop what they were not expected to report and might not perform according to the kind of development they were expected to show. However, before any hypothesis about indulgence and parental stress or engagement (Haine-Schlagel & Escobar-Walsh, 2015), is necessary to know if the CBI and mediating sociodemographic factors might change children’s outcomes, as other parental interventions has been shown to do (Biel et al., 2020).

In the discussion, p. 31-32:

The microsystems dynamics related with the CBI had focused on promoting engagement with sensitive and responsible parenting, but indicated distinct barriers (e.g., employment challenges, health-related challenges) and facilitators (e.g., knowing other mothers in the group, interest in the program topics) had a significant effect in recent studies with socioeconomically disadvantaged families (So et al., 2020), none of which were explored in the current CBI study. Also, CBI is an intervention for child development supported by current models of early childhood interventions recommended in scalable programs to impact early development (Cavallera et al., 2019; Pérez-Escamilla, Cavallera, Tomlinson & Dua, 2018), specifically in dedicated caregiving from parents (Martins et al., 2020) and low-cost interventions, contrary to exclusively use home visits. The emphasis in remote CBI compared with exclusively home visits interventions is not only monetary: high parental engagement with home visits is associated with particular maternal characteristics (higher IQ, older mothers, mothers who were employed during pregnancy, mothers with greater knowledge of infant development, and mothers with more positive parenting beliefs), rather than other characteristics (i.e., young, unemployed, and/or less well-educated mothers) that are more frequently found in LMICs (Doyle, 2020).

Overall

11. A minor concern relates to clarity and grammaticality throughout the manuscript. The article would benefit from a thorough review.

Reply/action: A revision was made before resubmission by a native English speaker, a highly qualified University associated professor living in Colombia. 

6. PLOS authors have the option to publish the peer review history of their article (what does this mean?). If published, this will include your full peer review and any attached files.

Do you want your identity to be public for this peer review? For information about this choice, including consent withdrawal, please see our Privacy Policy.

Reviewer #1: No

---

## [Decision Letter · Decision Letter 1]

22 Mar 2023

PONE-D-21-40928R1Parental developmental screening with CARE: A pilot remote assessment and intervention with vulnerable families in ColombiaPLOS ONE

Dear Dr. Giraldo-Huertas,

Thank you for submitting your manuscript to PLOS ONE. We feel your work has merit, and we appreciate your work to address reviewers' concerns. However, we feel a few additional changes are needed to fully meet PLOS ONE’s publication criteria. Therefore, we invite you to submit a revised version of the manuscript that addresses the points raised during the review process.

 I and one additional reviewer have now had the opportunity to review your manuscript. My review aligns with Reviewer 2's, particularly in regard to the need to clarify which parts of the intervention require internet or phone access and which require in-person visits, as well as additional clarification around analyses and outcomes. You can see Reviewer 2's full comments below. We look forward to reading and reviewing your revised manuscript!

We look forward to receiving your revised manuscript.

Kind regards,

Caitlin F. Canfield

Guest Editor

PLOS ONE

Journal Requirements:

Reviewers' comments:

Reviewer's Responses to Questions

**Comments to the Author**

1. If the authors have adequately addressed your comments raised in a previous round of review and you feel that this manuscript is now acceptable for publication, you may indicate that here to bypass the “Comments to the Author” section, enter your conflict of interest statement in the “Confidential to Editor” section, and submit your "Accept" recommendation.

Reviewer #2: (No Response)

2. Is the manuscript technically sound, and do the data support the conclusions?

Reviewer #2: Yes

3. Has the statistical analysis been performed appropriately and rigorously? 

Reviewer #2: Yes

4. Have the authors made all data underlying the findings in their manuscript fully available?

Reviewer #2: No

5. Is the manuscript presented in an intelligible fashion and written in standard English?

Reviewer #2: Yes

6. Review Comments to the Author

Reviewer #2: The present study sought to examine the utility of a primarily remote and parent-focused intervention aiming to improve childhood developmental outcomes. The study additionally focused in particular on socioeconomically vulnerable families living in Columbia. A quasi-randomised controlled trial design was utilized in order to investigate the impact of this intervention, as well as the effect of various sociodemographic variables, on outcomes of interest. The authors conclude that, compared to the control group, children of parents who participated in the intervention showed improvements in overall developmental delays. Further, it was revealed that sociodemographic variables such as socioeconomic status were also found to have moderating effects on outcomes. This manuscript makes important contribution in providing methodologically sound empirical support for an intervention aiming to improve childhood outcomes in a vulnerable population. However, there are a few concerns, including notable concerns for grammar and clarity, outlined below in no particular order. I hope the authors find them helpful.

Major Comments

1. In all, clarification of which parts of the intervention require internet access, which require in person appearances, and which are entirely remote, is needed. The authors state that remote interventions are needed in order to allow wider access (i.e., access for those not able to attend in person and/or without internet or phone access). However, it appears that the present intervention requires some phone or internet access (e.g., email) as well as some in-person sessions (e.g., intervention meetings and parent small group meetings). Explicit clarification on format for different parts of the intervention is needed. Given multiple formats utilized, the intervention may be better portrayed as hybrid, than fully remote.

Minor Comments

1. Given the study’s focus on the benefits of intervention, it is recommended the study is referred to as an efficacy study as opposed to a validation study

2. In the introduction the authors note “The selection criteria for the HLL are intended to increase the rigor in items for observation in Latin American Spanish…” Is the goal to state that this is the reason the authors have selected this measure, or that this is the rationale for the measures development? Please clarify?

3. Authors state “Results of a receiver operating curve (ROC) analysis indicated that parental report using CARE is a satisfactory tool for screening diagnostics (Area Under the Curve - AUC: 0.894, trapezoidal Wilcoxon area)..” Are these analyses done as a part of the present study? If no, please cite. If yes, please include these in results section.

4. Are secondary outcomes, e.g., literacy skills, assessed in English or Spanish? This is currently not specified and clarification would be helpful.

5. In the methods section, a brief discussion of reasons respondees declined participation if known, is warranted. If known, please also discuss how this may have impacted results in the discussion section. Figure 1 summarizing screening flow also needs to be edited for grammar (e.g., ‘Mother get employment,’ ‘we want to received the books’).

6. Page 14, gender should also be listed as a demographic factor for which no group difference was found.

7. It is noted on page 17 that caregivers’ accounts were frequently inconsistent. A discussion of impact of this observation on outcomes is recommended.

8. Clarification of the analysis done leading to statement that “primary outcomes presented positive improvements,” page 22, should be added.

9. P-values should be added to table 4. Presenting results of primary and secondary outcomes in table format, such as that of table 4, may be helpful.

10. Limitations and future research section in particular should be edited for grammar and clarify (e.g., sentence: A formal comparison between our parent-based intervention and a center-based intervention like that reported by Nores et al. requires a comprehensive analysis of the scientific literature relating to comparisons of interventions in Spanish-speaking and LMIC conditions; and so far, there has been no such study).

The following typos are noted:

1. “Also, thr HLL was selected” (page 7)

2. “The COVID-19 pandemic had causal effects on mental health and caregiving environments (Pitchik et al., 2021), exacerbated in contexts of poverty, which increasing the risk of children not reaching their developmental potential in Colombia)” (page 31)

3. Missing period after (Martins et al., 2020) (page 32)

7. PLOS authors have the option to publish the peer review history of their article (what does this mean?). If published, this will include your full peer review and any attached files.

Reviewer #2: No

---

## [Author Response · Author response to Decision Letter 1]

1 May 2023

PONE-D-21-40928R1

Parental developmental screening with CARE: A pilot remote assessment and intervention with vulnerable families in Colombia

PLOS ONE

Dear Reviewers, I am very grateful for the opportunity to respond to your review of my manuscript. The full replies are below. 

Best regards,

Juan J Giraldo-Huertas, PhD

To Major Comments: 1. In all, clarification of which parts of the intervention require internet access, which require in person appearances, and which are entirely remote, is needed. The authors state that remote interventions are needed in order to allow wider access (i.e., access for those not able to attend in person and/or without internet or phone access). However, it appears that the present intervention requires some phone or internet access (e.g., email) as well as some in-person sessions (e.g., intervention meetings and parent small group meetings). Explicit clarification on format for different parts of the intervention is needed. Given multiple formats utilized, the intervention may be better portrayed as hybrid, than fully remote.

R/ more precision in the content of the Method Section and specifical indications about the modality of participation are included in Table 2 (p. 18).

To Minor Comments:

1. Given the study’s focus on the benefits of intervention, it is recommended the study is referred to as an efficacy study as opposed to a validation study.

R/ Corrected in abstract (p. 2)

2. In the introduction the authors note “The selection criteria for the HLL are intended to increase the rigor in items for observation in Latin American Spanish…” Is the goal to state that this is the reason the authors have selected this measure, or that this is the rationale for the measures development? Please clarify?

R/Corrected in p. 7:

The selection criteria for the HLL serve two purposes. Firstly, they are intended to allow for easy and rigorous adaptation of developmental screening items in Latin American Spanish through cultural and linguistic modifications obtained from the original Llevant study in the Basque Country (Fuentes-Biggi, Fernandez, & Alvarez, 1992). Secondly, the HLL is a widely-used and standardized adaptation of the DDST in various countries, including densely populated areas in Brazil (Lopez-Boo, Cubides-Mateus, & Llonch-Sabatés, 2020) and Colombia (Rubio-Codina & Grantham-McGregor, 2020).

3. Authors state “Results of a receiver operating curve (ROC) analysis indicated that parental report using CARE is a satisfactory tool for screening diagnostics (Area Under the Curve - AUC: 0.894, trapezoidal Wilcoxon area)..” Are these analyses done as a part of the present study? If no, please cite. If yes, please include these in results 

section.

R/ Corrected in p. 7:

All the reliability, agreement and diagnostic-characteristics analysis results were previously carried and published before the intervention and efficacy study (Giraldo-Huertas & Schafer, 2021).

4. Are secondary outcomes, e.g., literacy skills, assessed in English or Spanish? This is currently not specified and clarification would be helpful.

R/ Corrected in p. 19:

The outcomes were assessed exclusively in Spanish at a post-intervention assessment conducted 7-8 weeks following the baseline assessment. The assessments were conducted in-person at the Children’s Centre by two native Spanish-speaking graduate psychologists who were blind to group membership and trained in the specific measures used. The full set of assessments took 60-90 minutes. The following primary and secondary outcomes were administered at baseline and follow-up using the official or standardized version in Spanish.

5. In the methods section, a brief discussion of reasons respondees declined participation if known, is warranted. If known, please also discuss how this may have impacted results in the discussion section. 

R/ Corrected in p.12 and p. 36.

The baseline assessment was conducted if the child met the entry criteria and the parents consented to participate. Prior to the COVID-19 pandemic (July 2019 – February 2020), the screening process identified 268 families (see Fig 1 for the CONSORT flowchart), but 156 families (54.9%) declined to enter the study and were not followed up further. The reasons for their decision were not collected.(p. 12)

The present study has highlighted significant practical and methodological challenges in conducting an RCT of a remote intervention for parents in low-income conditions with preschool children at risk of not reaching their developmental potential. One major obstacle in our study was the high rate of declining to join after the initial call, which affected our recruitment rate - more than half of the potential families who could have received the intervention (54.9%) opted not to participate. Unfortunately, the reasons for their decision were not collected. Recent meta-analyses and evidence-based studies (Kumar et al., 2022) report that failure to inform on attrition rates and reasons for attrition can impact the integrity of interventions in LMICs, even in home visit intervention designs (Bai, Abulitifu, & Wang, 2022). To ensure the quality of future studies, researchers should not only collect information about attrition and non-participation but also incorporate principles and practices that align with new forms of services offered in public health initiatives, such as family-focused and community-based approaches (McConkey, 2022). (p. 36)

In this correction, the following references are included:

Bai, Y., Abulitifu, R., & Wang, D. (2022). Impact of an Early Childhood Development Intervention on the Mental Health of Female Caregivers: Evidence from a Randomized Controlled Trial. International journal of environmental research and public health, 19(18), 11392. https://doi.org/10.3390/ijerph191811392

Kumar, S., Narayan, S., Malo, P., Bhaskarapillai, B., Thippeswamy, H., Desai, G., & Kishore, M. (2022). A systematic review and meta-analysis of early childhood intervention programs for developmental difficulties in low-and-middle-income countries. Asian journal of psychiatry, 70, 103026. https://doi.org/10.1016/j.ajp.2022.103026

McConkey R. (2022). Responding to Autism in Low and Middle Income Countries (LMIC): What to Do and What Not to Do. Brain sciences, 12(11), 1475. https://doi.org/10.3390/brainsci12111475

6. Figure 1 summarizing screening flow also needs to be edited for grammar (e.g., ‘Mother get employment,’ ‘we want to received the books’).

R/ Corrected in Figure 1: “Mother get a full-time job” and “We want to receive the books”.

7. Page 14, gender should also be listed as a demographic factor for which no group difference was found.

R/ Corrected in p.14

8. It is noted on page 17 that caregivers’ accounts were frequently inconsistent. A discussion of impact of this observation on outcomes is recommended.

R/ Corrected in p. 17 and p. 32

At the second session (see Table 2), and each subsequent remote session, the facilitator checked via phone call, chat, or email message if that week’s daily records had been used, and if any difficulties had been encountered. It was not possible to determine any significant changes week by week and include in the results analysis the daily reporting of activities at home because caregivers’ written, and verbal accounts were frequently inconsistent in the registration area of each booklet. (p. 17)

However, the consistency and quality of the daily-based information recorded in the CARE booklet were problematic, which made it challenging to identify significant changes in microsystems level by time. Although this is a common issue in research that relies on self-reported data, it is important to acknowledge and address potential sources of bias or errors in the data collection process. In future studies using the CARE booklet, alternative data collection methods or strategies to improve the quality and consistency of the recorded information may prove useful. (p. 32)

9. Clarification of the analysis done leading to statement that “primary outcomes presented positive improvements,” page 22, should be added.

R/ Statement corrected in p.22: “primary outcomes presented positive score changes in the CBI group”.

10. P-values should be added to table 4. Presenting results of primary and secondary outcomes in table format, such as that of table 4, may be helpful.

R/ Data of P-values included in Table 4 (p. 26)

11. Limitations and future research section in particular should be edited for grammar and clarify (e.g., sentence: A formal comparison between our parent-based intervention and a center-based intervention like that reported by Nores et al. requires a comprehensive analysis of the scientific literature relating to comparisons of interventions in Spanish-speaking and LMIC conditions; and so far, there has been no such study).

R/ Corrected from p.33 to p.37

All effect sizes in a pilot study should be interpreted with caution because there are several limitations due to the small sample size. A larger sample size is needed for a rigorous and detailed mapping of the effects of similar procedures on poor, low-income families in LMICs, including adequate controls for sociodemographic variation for future post hoc analysis after the two-way ANCOVA procedures to properly evaluate the CBI. 

Comparing the CBI with Colombian center-based interventions like the aeioTu program (Nores et al., 2019) might not be appropriate because it requires a control group with no local service allocation. However, the CBI is a suitable alternative to consider when comparing the child-to-adult dyadic ratio and the length of the intervention. To make a formal comparison between our parent-based intervention and a center-based intervention like that reported by Nores et al., a comprehensive analysis of the scientific literature relating to interventions in Spanish-speaking and LMIC conditions is necessary, but no such study has been conducted so far.

Families allocated to the control (local services only) group were assigned to the control condition because they did not present themselves on the day of randomized allocation and cannot be considered a truly random control group. However, this reflects a real-life clinical and research situation. Future RCT studies should be completely coherent with randomized allocation and sufficient sample size to avoid potential biases and to increase statistical power to generalize the differences between groups detected. 

Another limitation of our study is related to our measure of development. The Haizea-Llevant observation table (HLL) is meant as a screening tool and not a diagnostic tool. A high number of Delay and Caution items in the HLL developmental dimensions should not be strictly interpreted as a delay relative to benchmarks in comparison with children of the same age. Therefore, the results were framed in terms of children being "at risk" of a loss of developmental potential, rather than as a diagnosis of a specific developmental delay. 

The generalizability of our findings may be affected by using a sample from a community-based services program. Participation in the nutritional program offered in the participating childcare center was voluntary, and parents who had concerns about their child's development may have been more likely to stay in the program than parents who found their children to be on track. This could have inflated our estimates of the prevalence of risk for developmental delay. However, our sample had relatively similar proportions of various sociodemographic risk factors as the broader group of participants previously seen as at risk of developmental delay (Edmunds, 2020; Richter et al., 2019), and for these reasons, it seems reasonable to assume that our sample does not substantially overestimate the risk for developmental delay in this low-income LMIC population.

A major limitation of this study is the lack of direct observational assessment of parent skills, such as caregiver social play sensitivity, which has been shown to produce significant positive changes in infants’ cognitive and socioemotional outcomes in previous parental and language interventions (Holmberg et al., 2022; Murray et al., 2016). The absence of such assessment was due to scarce financial resources for the Q-RCT. Additionally, levels of commitment of parental involvement (CPI) were not assessed in previous stages of recruitment. A meta-analysis by Haine-Schlagel and Escobar-Walsh (2015) reported that most studies indicate how parents face environmental and personal challenges to participate actively and concluded that levels of engagement are better understood when including a consideration of parental involvement issue. Measures of CPI fall into three main categories: global participation levels, specific participation behaviors, and completion of tasks. The average completion rate for all sessions analyzed by Haine-Schlagel and Escobar-Walsh was 49%, with a range of 19% to 89%. One possible explanation for the low engagement in optional activities for the present study (54.9% at first call) could be attributed to the sociodemographic characteristics that influenced the opportunity of adherence for participants. Comparing our sample and experimental procedure with an RCT of a play-assisted intervention for children living with foster families in extreme poverty (Worku et al., 2018), two characteristics could affect the CPI in our case. First, families tended not to cancel visits in the Worku et al. intervention. In contrast, our families received all assessments by arriving at the childcare center (CC) and interventions via chat or email message at home. Second, Worku and colleagues’ (2018) intervention was conducted in a foster family program (SOS-villages), where children lived with assigned foster families and were always cared for by an SOS-village mother or “aunt”. In future, direct measures of parental behavior and competence need to be included to demonstrate objective, generalizable difficulties in CPI. 

The present study has highlighted significant practical and methodological challenges in conducting an RCT of a remote intervention for parents in low-income conditions with preschool children at risk of not reaching their developmental potential. One major obstacle in our study was the high rate of declining to join after the initial call, which affected our recruitment rate - more than half of the potential families who could have received the intervention (54.9%) opted not to participate. Unfortunately, the reasons for their decision were not collected. Recent meta-analyses and evidence-based studies (Kumar et al., 2022) report that failure to inform on attrition rates and reasons for attrition can impact the integrity of interventions in LMICs, even in home visit intervention designs (Bai, Abulitifu, & Wang, 2022). To ensure the quality of future studies, researchers should not only collect information about attrition and non-participation but also incorporate principles and practices that align with new forms of services offered in public health initiatives, such as family-focused and community-based approaches (McConkey, 2022). 

Moreover, future studies aiming to improve interaction skills between caregivers and children (Biel et al., 2020) may consider using a low-cost and widely applicable booklet, such as the CARE booklet intervention, to guide parental scaffolding. Our general findings on various developmental screening and language competence variables suggest the need for further work on approaches to parental training. This should include RCTs with sufficient sample size and methodological rigor to confirm and extend these tentative findings, and to more clearly demonstrate whether parent training with scaffolding guides has specific beneficial effects on relevant skills and competences of children at risk. One type of intervention includes screening or early developmental monitoring (Cavallera et al., 2019; Goldfeld & Yousafzai, 2018) to accomplish remote assessment and intervention in the most vulnerable populations of LMIC.

In this point, the following reference is included:

Holmberg, E., Kataja, E. L., Davis, E. P., Pajulo, M., Nolvi, S., Hakanen, H., Karlsson, L., Karlsson, H., & Korja, R. (2022). The Connection and Development of Unpredictability and Sensitivity in Maternal Care Across Early Childhood. Frontiers in psychology, 13, 803047. https://doi.org/10.3389/fpsyg.2022.803047

12. The following typos are noted:

1. “Also, thr HLL was selected” (page 7) – Corrected.

2. “The COVID-19 pandemic had causal effects on mental health and caregiving environments (Pitchik et al., 2021), exacerbated in contexts of poverty, which increasing the risk of children not reaching their developmental potential in Colombia)” (page 31) – Corrected:

The COVID-19 pandemic had causal effects on mental health and caregiving environments, particularly in contexts of poverty. These effects have been observed in Colombia and other LMICs, and have increased the risk of children not reaching their developmental potential (Pitchik et al., 2021).

3. Missing period after (Martins et al., 2020) (page 32) – Corrected.

---

## [Editor Report · Decision Letter 2]

1 Jun 2023

Parental developmental screening with CARE: A pilot hybrid assessment and intervention with vulnerable families in Colombia

PONE-D-21-40928R2

Dear Dr. Giraldo-Huertas,

We’re pleased to inform you that your manuscript has been judged scientifically suitable for publication and will be formally accepted for publication once it meets all outstanding technical requirements.

Kind regards,

Shivanand Kattimani

Academic Editor

PLOS ONE

**Additional Editor Comments:**

Dear authors, thank your submitting your manuscript to us.

Following are my specific comments

Comment 01: Title is bit confusing. Now it is very dry and difficult to engage readers. I suggest following alternative titles. However, decision is yours.

1. Developmental screening with CARE booklet as Intervention by parents in Preschoolers: effect on general development, literary, reading and narrative skills

2. Developmental screening with CARE booklet as Intervention by parents in Preschoolers from vulnerable families in Colombia: effect on general development, literary and language skills

3.Parents from vulnerable families using Developmental screening CARE booklet as intervention in Preschool children: A pilot hybrid efficacy study

4.Development screen using CARE booklet as intervention by vulnerable families in Colombia in preschoolers: Impact on general development and literary-language skills

Coment 02: Mention in the abstract.

target sample was preschoolers with age range, and duration of intervention (gap between baseline intervention and assessement of outcomes)

comment 03: mention if anybody else qualifies for authorhsip or for being acknowldged for contribution in the study

---

## [Editor Report · Acceptance letter]

20 Jun 2023

PONE-D-21-40928R2 

Parental developmental screening with CARE: A pilot hybrid assessment and intervention with vulnerable families in Colombia 

Dear Dr. Giraldo-Huertas:

I'm pleased to inform you that your manuscript has been deemed suitable for publication in PLOS ONE. Congratulations! Your manuscript is now with our production department. 

Kind regards, 

on behalf of

Dr. Shivanand Kattimani 

Academic Editor

PLOS ONE